# Oncogenic KRAS: Signaling and Drug Resistance

**DOI:** 10.3390/cancers13225599

**Published:** 2021-11-09

**Authors:** Hyeon Jin Kim, Han Na Lee, Mi Suk Jeong, Se Bok Jang

**Affiliations:** 1Department of Molecular Biology, College of Natural Sciences, Pusan National University, Jangjeon-dong, Geumjeong-gu, Busan 46241, Korea; khjkhj0903@naver.com (H.J.K.); emille96@naver.com (H.N.L.); 2Institute for Plastic Information and Energy Materials and Sustainable Utilization of Photovoltaic Energy Research Center, Pusan National University, Jangjeon-dong, Geumjeong-gu, Busan 46241, Korea

**Keywords:** KRAS, GTPase, signaling, mutant, inhibitor, drug resistance

## Abstract

**Simple Summary:**

KRAS is one of the oncogenic proteins and best well-known GTPase proteins. Although KRAS serves as regulation of cellular process, abnormal mutated and GTP-bound KRAS is highly expressed in human cancers. Understanding biological characteristics and precise signaling of KRAS proteins is essential for targeted cancer therapy. This review explains signal transduction and exploration of targeted anticancer by activated KRAS.

**Abstract:**

RAS proteins play a role in many physiological signals transduction processes, including cell growth, division, and survival. The Ras protein has amino acids 188-189 and functions as GTPase. These proteins are switch molecules that cycle between inactive GDP-bound and active GTP-bound by guanine nucleotide exchange factors (GEFs). KRAS is one of the Ras superfamily isoforms (N-RAS, H-RAS, and K-RAS) that frequently mutate in cancer. The mutation of KRAS is essentially performing the transformation in humans. Since most RAS proteins belong to GTPase, mutated and GTP-bound active RAS is found in many cancers. Despite KRAS being an important molecule in mostly human cancer, including pancreatic and breast, numerous efforts in years past have persisted in cancer therapy targeting KRAS mutant. This review summarizes the biological characteristics of these proteins and the recent progress in the exploration of KRAS-targeted anticancer, leading to new insight.

## 1. KRAS Protein

### KRAS Structure, Function

KRAS is one of the RAS superfamilies or Ras-like GTPase and belongs to the group of small-guanosine triphosphate (GTP) binding proteins. The Ras protein is one of the superfamilies that shares functional small GTPases and is composed of RAS, RHO, RAB, RAC, RAN, and ARF (Figure 1) [1]. The RAS family is divided into six subfamilies (RAS, RAL, RAP, RHEB, RAD, and RIT). The RAS family commonly has a catalytic G domain, which generates GTPase. Also frequently studied in the RAS family are a Harvey-Ras (H-RAS), neuroblastoma-Ras (N-RAS), and two variants of Kristen-RAS (K-RAS). While the first variant KRAS4A is weakly expressed in the human cell, the second variant KRAS4B is excessively expressed [2,3]. KRAS protein contains 188 amino acids, with a molecular mass of 23.2kDa, and is performed as the first sensor of intracellular signaling pathway [2,4]. Activation of KRAS is regulated by the on-off switch, resulting in an intracellular effector pathway. The ‘on’ and ‘off’ conformation is proceeded by binding of GTP and GDP (Figure 2). Under physiological conditions, the transition between GDP and GTP is involved in GEF by promoting GDP for GTP exchange and GAP by accelerating RAS-mediated GTP hydrolysis [2,5]. KRAS protein-bounded GTP is undergoing conformation change at two regions. The first region (amino acids 30–38) and the second region (amino acids 59–67) form an effector loop, controlling the binding of GTPase to effector molecules. Once conformational change occurs, the effector molecules of GTP activity, such as GTPase-activating proteins (GAPs) and guanine-exchange/releasing factors (GEFs/GRFs), affect the interaction [2]. The GAPs are able to amplify the GTPase activity of KRAS protein 100,000-fold more than RAS protein-bounded GDP [6]. Contrariwise to activation of GTP activity, the GEFs/GRFs reduce GTPase activity by promoting the release of GTP [2]. Thus, the mutation of KRAS protein impairs GTPase activity, which is mainly found in the oncogene RAS form. The mutation of RAS p21 tends to rapidly exchange GDP for GTP, which induces an active form of RAS. An aberrant form of KRAS protein affects the effector that is related to the important cellular pathway. Accordingly, hyperactivation of RAS signaling is related to cancer and human disease such as RASopathies [7].

RAS has three domains (Figure 3). RAS isoforms share G-domain that highly sequences homology (90%). The G-domain (residues 1–166) consists of an effector lobe (residues 1–86) and an allosteric lobe (87–166). This domain is composed of the six beta-strands and is surrounded by five alpha-helices [9]. KRAS has hypervariable regions (HVRs) present in the C-terminal element. The HVRs region undergoes a post-translational modification (PTM), including by farnesylation, palmitoylation, methylation, phosphorylation, nitrosylation, and ubiquitination [10]. Particularly, the carboxyl terminal PTMs are related to affinity for membrane and trafficking with cellular compartments. According to research, palmitoylated HRAS is located in the plasma membrane, but depalmitoylated HRAS is accumulated in the ER and Golgi [11]. The HVRs play a role in anchoring RAS to the membrane, which displays among Ras proteins less than 15% of sequence homology [12]. The effector lobe mainly interacts with RAS effectors such as rapidly accelerated fibrosarcoma (RAF), PI3K, and RalGEF. The allosteric lobe plays a role in intra-protein communication, which connects the effector lobe to membrane-interacting residues [13,14]. The effector lobe is divided into two switch regions, switch-I (residues 30–38) and switch-II (residues 59–76), controlling the specificity of the binding to effector molecule and affecting interaction of transducer to cellular downstream [2,7]. Different from the Ras protein, the effector lobe of KRAS additionally has P-loop-phosphate binding loops (residues 10–16 and 56–59), which indicate a GTP-binding pocket. These play a role in interacting with the b-phosphate and c-phosphate of GTP [2]. Moreover, residues 116–119 and 152–165 interact with the guanine base. In the case of binding of GTP with P-loop-phosphate, GTP temporarily binds to Ras protein. Additionally, KRAS protein has a CAXX motif, which is related to plasma membrane localization. The KRAS protein should localize plasma membrane to regulate signal transduction. For the plasma membrane localization, KRAS protein is needed for farnesylation in the cysteine residue of CAXX (C represents cysteine, A is an aliphatic amino acid, and X is any amino acid) motif [15,16]. After farnesylation of cysteine residues in the CAXX, the AXX amino acid is cleaved by protease, followed by the methylation the free carboxyl group of the cysteine, and these events occur on the cytosolic surface of the endoplasmic reticulum [16]. Additionally, in the case of the splice variant KRAS4A, the AXX motif is palmitoylated by palmitoyl transferase and targeted to the membrane. However, in the case of the splice variant KRAS4B, it is speculated that membrane localization occurs through a microtubule-dependent mechanism [17].

## 2. KRAS Signaling

In the cell membrane, KRAS undergoes inactive state by binding of guanosine diphosphate (GDP) and active state by binding of guanosine triphosphate (GTP). KRAS protein is normally maintained in oncogenesis, but mutated KRAS is decreased in interactions with GTPase activator protein [18]. Moreover, the mutated KRAS tends to prefer binding with GTP than GDP. The activation form of GTP-bound Ras protein influences the activation downstream effector even in the absence of growth factor [19]. Typically, the Ras signal pathway is activated by the protein tyrosine kinase receptor. The epidermal growth factor receptor (EGFR) and the platelet-derived growth factor receptor (PDGFR) are well-known protein tyrosine kinase receptors at the Ras signal pathway. For the Ras signal to be activated, the ligand binds the EGF receptor, followed by induces oligomerization of the receptor [20]. This can allow activation of the kinase activity and transphosphorylation of catalytic domains [20]. The sequence homology 2 (SH2) domain of the receptor is a recognized adaptor protein such as growth factor receptor-bound protein 2 (Grb2), which in turn recruits guanine nucleotide exchange factors (GEFs) like son of sevenless homolog 1 (SOS-1) to the cell membrane [19]. The GEFs are able to interact with Ras protein to promote a conformational change and exchange GDP for GTP. After activation of Ras protein, various downstream effectors such as canonical Raf/Mek/Erk, phosphatidylinositol 3-kinase (PI3K)/3-phosphoinositide-dependent protein kinase-1 (Pdk1)/Akt, and Ral-GEF are recruited (Figure 4) [5,21].

### 2.1. RAF-MEK-ERK Pathway

The first well-known Ras effector pathway is identified as the RAF-MEK-ERK pathway. This pathway is involved in the mitogenic signaling of tyrosine kinase receptors, following a wide range of growth, differentiation, inflammation, and apoptosis [22]. Raf is a member of the family of serine/tyrosine kinase. Serine/tyrosine kinase includes Raf-1, A-Raf, and B-Raf [19]. This family binds to the effector region of RAS-GTP, resulting in translocation of the Raf protein to the plasma membrane [23,24]. According to the research, Ras activated either by point mutation (G12V) or by GTP-bound interacts with Raf-1 through guanylyl-imidodiphosphate (GMP-PNP). However, effector domain mutation (Ile36Ala) of Ras does not interact with Raf-1 [25]. Following Ras interacting with Raf, Raf protein is activated and phosphorylated by protein kinase. Raf activation induces a signaling cascade, which phosphorylates mitogen-activated protein kinase (MAPK), which in turn, phosphorylates and activates the downstream effector such as extracellular signal-regulated kinase 1 and 2 (ERK1 and ERK2) [19]. Activation of ERK kinases plays a role in Ras-induced cellular responses and translocation to the nucleus [21]. Therefore, ERK activates and phosphorylates the nuclear transcription factor and kinases such as eukaryotic initiation factor 2alpha kinase 1 (EIK-1) and protein C-ets-1,2 (c-Ets1, c-Ets2) [19]. Besides activation of ERK, MAPK, also called MEK, is activated in signal response that promotes cell survival and apoptosis through various mediators such as c-Jun N-terminal kinases (JNK), stress-activated protein kinase (SAPK), and nuclear factor kappa-light-chain-enhancer of activated B cells (NF-KB) [26].

In cancer, studies targeting Raf kinase have been demonstrated. According to the research for the influence of RAF-MEK-ERK pathway in tumor growth, expressing dominant-negative MEK in tumor cells affects a reduction in endogenous Raf-1 and MEK-1 activity [27]. Moreover, research demonstrated that clones expressing the dominant-negative MEK form the smallest tumors and appear to increase survival [27]. Phosphorylation of MEK occurs in residue serine 218 and 222, but MEK point mutation in which these serines are substituted by alanine (S218A/S222A) is not able to be activated Raf [28]. Additionally, inhibition of the RAF-MEK-ERK pathway via mutant MEK demonstrates reversion of the cancer phenotype, and inhibitors of Raf kinase are targeted anti-tumor therapies [29].

### 2.2. PI3K Pathway

The second well-known Ras effector family is phosphoinositide 3-kinases (PI3Ks). PI3K plays a role in mediators of Ras-mediated cell survival and proliferation [30]. Activation of PI3K converts phosphatidylinositol (4,5)-biphosphate (${\mathrm{PIP}}_{2}$) into phosphatidylinositol (3,4,5)-triphosphate (${\mathrm{PIP}}_{3}$). ${\mathrm{PIP}}_{3}$, in turn, binds the pleckstrin homology (PH) domain of Akt/PKB. Akt/PKB stimulates phosphorylation of proteins that affect cell growth, cell cycle, and survival [21]. To activate PI3K, binding of a ligand to receptor tyrosine kinase (RTK) first occurs. Following binding of the ligand in RTK, RTK is dimerized and autophosphorylated, leading to interaction with PI3K effector via Src homology 2 (SH2) domains [31]. Adaptor protein of PI3K, GRB2 binds to phosphor-YXN motifs of the RTK. In turn, GRB2 binds and activates SOS, which then activates RAS. Activation of RAS activates p110 of p85 [32]. The tumor suppressor phosphatase and tensing homology (PTEN) dephosphorylates ${\mathrm{PIP}}_{3}$ to ${\mathrm{PIP}}_{2}$. ${\mathrm{PIP}}_{3}$ induces intracellular signaling via binding to protein with PH domains, such as phosphoinositide-dependent kinase (PDK1) and AKT [33]. The PI3K-AKT pathway promotes cell growth and survival [21,34].

The importance of PI3K in the RAS-signaling pathway is highly correlated with oncogenic signaling. According to the research, there is evidence that studies have found somatic missense mutations in PIK3CA in human tumors, including those of the brain, breast, and colon [35,36,37].

### 2.3. Ral-GEF Protein

The Ral-GEF protein also became known as the Ras effector. Ral-GEF members are composed of RalGDS, RGL, RGL2, and RGL3 [38]. The members are activated the RalA and RalB small GTPase via the link with Ras proteins. Thus, the RalGEF protein is a small GTPase that provides GTP to the Ras superfamily. According to the research, GTP-bound RalA and RalB are discovered to have high levels in human pancreatic ductal adenocarcinoma (PDAC). Moreover, inhibition of RalA expression reduces tumorigenic growth [39].

## 3. RAS Inhibitors and Resistance

Acquired resistance occurs for several reasons. Pre-existing and de novo mutations provide chemoresistance and acquired resistance [40]. Resistance mechanisms happen in a very rare subpopulation of cells, such as cancer stem cells [41,42]. Kumar, B. et al. revealed that enhanced autophagy induces cancer cell stemness and promotes resistance to colorectal cancer therapy. Moreover, 36-077 promotes the efficacy of colorectal cancer therapy. 36-077 inhibits phosphatidylinositol 3-kinase (PIK3C3/VPS34 kinase) signaling, a key regulator of autophagy, and a potent target for inhibiting autophagy [43]. Long noncoding RNA (lncRNA) is highly expressed in cancer stem cells. lncRNA HAND2-AS1 is required for self-renewal of cancer stem cells, through activation of the bone morphogenetic protein (BMP) signaling [44,45]. Taken together, many pathways and associated factors play a key role in mediating drug resistance.

### 3.1. Inhibiting the Post-Translation Processing of RAS

RAS gene encodes three genes, including HRAS, NRAS, and KRAS. KRAS has two alternatively spliced mRNA variants, KRAS4A and KRAS4B [46]. These proteins with amino-terminal residues 1–165 share 92~98% sequence identity [47]. RAS proteins carry 23~24 carboxyl-terminal residues, including the CAAX box. CAAX box undergoes four-stage post-translational lipid modification: (1) farnesylation by FTase, (2) prenylation at the CAAX cysteine residue followed by proteolytic [42] removal of the terminal -AAX by Rac converting CAXX endopeptidase 1 (RCE1), (3) carboxymethylation of the C-terminal cysteine by isoprenylcysteine carboxyl methyltransferase (ICMT), and (4) palmitoylation of cysteine residues located upstream of the C-terminus of the protein [48,49].

The excision of RCE1 eliminated RAS endoproteolytic processing and mislocalization of RAS proteins. Moreover, the loss of RCE1 significantly reduces the growth of cells in the skin carcinoma cell line [50]. Treatment of cysmethynil, ICMT inhibitor, results in mislocalization of RAS and inhibition of cell growth and KRAS-induced oncogenic transformation in nude mice [51] (Table 1). TLN-4601 has been shown to disrupt RAS membrane anchorage and anti-tumor activity in several xenograft models. Interestingly, the compound induces Raf-1 proteasomal-dependent degradation and may inhibit the MEK/EKR pathway by depleting the Raf-1 protein [52]. Liu et al. demonstrated that inhibitors of FTase and geranylgeranyltransferase-I were shown to reduce tumor growth and improved survival in mice with K-RAS-induced lung cancer [53]. Waldmann and colleagues targeted the small-molecule inhibitor deltarasin that interferes with the binding of mammalian phosphodiesterase δ (PDEδ) to KRAS and suppresses KRAS localization to endomembrane [54].

### 3.2. K-RAS Targeted Drug

The most mutation sites of oncogenic mutation in RAS are residues 12 and 13 in the P-loop, and residue 61 in switch II [72]. KRAS, which is most frequently mutated, contains 86% of RAS mutation. KRAS mutations occur in pancreatic, colorectal, and lung cancers. The vast majority of hotspot positions are on 12. NRAS mutations occur in melanoma and myeloid leukemia, with the most common site of mutation being residue 61. HRAS mutations occur with the highest frequency in bladder cancer, and hotspots are 12 and 61 residues [73] (Table 2).

ARS-583 reacts with the GDP-bound state of KRAS G12C. ARS-583 inhibits downstream MAPK/PI3K signaling across the group of cell lines and directly inhibits the treatment of patients with the KRAS G12C mutation, which comprises 20% of lung cancers [55]. ARS-1620 selectively and with high potency targets the switch II pocket in KRAS G12C. ARS-1620 achieves anti-tumor activity in subcutaneous xenograft models bearing KRAS G12C mutation [56]. Ryan et al. observed rapid adaptive RAS pathway feedback reactivation of MAPK signaling after treatment with ARS-1620 in KRAS G12C-driven lung, pancreatic, and colon cancer cells, suggesting secondary resistance via upregulated RTK signaling [74]. Treatment with AMG 510 regresses in KRAS G12C mouse xenografts and induces pro-inflammatory tumor microenvironment [57]. A covalent KRAS G12C inhibitor, MRTX849, selectively modifies mutant cysteine 12 in a GDP-bound state. MRTX849 has tumor regression in KRAS G12C positive cell lines and patient-derived xenograft models from multiple tumor types [58]. Small-molecule SOS1 inhibitor, BI-3406, interacts with the catalytic domain of SOS, resulting in interference with the interaction with KRAS and thereby restricting tumor cell proliferation in KRAS-mutant cancer, including G12 and G13 [59]. BI 1701963, which is a BI-3406 analog, is currently in a phase I trial, testing preliminary efficacy of BI 1701963 alone and in combination with trametinib in patients with KRAS mutant solid tumors [75]. Compounds of ponatinib and AMG-47a selectively reduce the levels of KRAS G12V proteins [60].

### 3.3. Inhibition of RAS Signaling Networks

Upstream RAS-mediated pathways have a common mechanism of resistance to tyrosine kinase inhibitors (TKIs) such as those targeting RTKs, including EGFR and FMS-like receptor tyrosine kinase-3 (FLT3) [40]. Treatment with anti-EGFR monoclonal antibodies cetuximab or panitumumab is successful in a subset of patients with colorectal cancer, although these stimulate MAPK reactivation, driving secondary resistance [76]. Resistance to EGFR inhibitor WZ4002 can induce reactivation of ERK1/2 signaling. Ercan et al. found that the combination of WZ4002 and MEK inhibitor may be an effective strategy to treat drug-resistant cancers [61]. EGFR inhibitor gefitinib showed drug resistance via Src-mediated ERK reactivation. Ochi et al. demonstrated that a combination of gefitinib and Src inhibitor may increase the potency of the RAS-inhibitor resistance mechanism to overcome this resistance [62]. Many different FLT3 inhibitors including gilteritinib, crenolanib, and midostaurin treat varying stages. Gilteritinib and midostaurin are FDA-approved whereas crenolanib is in phase II trials [63,64,65]. However, gilteritinib develops secondary resistance via activating RAS/MAPK pathway signaling in acute myeloid leukemia, most commonly in NRAS or KRAS [77]. The MRAS-SHOC2-PPI complex acts as a key role in the RAF-ERK pathway. Jones et al. reported that inhibition of SHOC2 inhibits tumor development in murine KRAS-driven lung cancer models. However, this compound activates other RAS-dependent pathways such as PI3K-AKT signaling, resulting in cancer that can persist [40,78]. Adachi and colleagues observed that epithelial-to-mesenchymal transition (EMT) led to intrinsic and acquired resistance by activation of PI3K signaling in the presence of KRAS G12C inhibitor. Treatment with the combination of KRAS G12C inhibitor, PI3K inhibitor, and SHP2 inhibitor resulted in tumor regression in mouse models of acquired resistance to KRAS G12C inhibitor, AMG510 [79].

### 3.4. KRAS Inhibitors Discovered by Virtual Screening

Covalent quinazoline-based switch II compounds effectively inhibit GTP loading of KRAS G12V, MAPK phosphorylation, and the growth of cancer cells driven by KRAS G12C [80]. Compound 23 (BAY-293) selectively inhibits the KRAS-SOS1 interaction and blocks reloading of KRAS with GTP. The compound suppresses the RAS-RAF-MEK-ERK pathway and inhibits the activation of RAS in tumor cells [66]. Fell et al. reported that tetrahydro pyridopyrimidines act as irreversible covalent inhibitors of KRAS G12C and can bind to the residue 12 and lock KRAS in its inactive GDP-bound form [67]. C19 bound and stabilized the KRAS4B-PDEδ complex. This compound suppresses the viability and proliferation of colorectal cancer cells and decreases tumor size in a xenograft mouse model via inhibiting phosphorylation of ERK and AKT signaling [68]. In addition, D14 and C22 stabilize the interaction with KRAS4B and PDEδ. D14 and C22 significantly decrease RAS-GTP activity and ERK and AKT pathways in pancreatic cancer cells [69].

### 3.5. Other Inhibitors Targeting RAS

CW Han et al. elucidated that there is direct interaction between KRAS G12V and H-REV 107 peptide with high affinity with the crystal structure. This peptide was shown to interact and stabilize with KRAS G12V inactive state resulting in the block of the activation function of KRAS. The peptide can inhibit pancreatic cancer and colon cancer cell lines in cell proliferation assay [70]. Treatment with platinum-based agents, such as cisplatin, carboplatin, and oxaliplatin, is used in a variety of cancers, including head and neck squamous cell carcinoma, testicular cancer, and non-small cell lung cancer [71,81,82]. Cisplatin induces mitochondrial ROS, which further increases DNA damage and induces increasing cell death. However, cisplatin shows an observed resistance mechanism, including the involvement of oncogenic KRAS mutations via NRF2 overexpressing [71].

## 4. Conclusions

RAS proteins are membrane-bound proteins and are frequently mutated in human cancers, with mutations in about 30% of all cancers. RAS exists in three ubiquitously expressed genes, HRAS, KRAS, and NRAS, with high sequence homology. RAS proteins cycle between an inactive (GDP-bound) and active (GTP-bound) state. When activated by a ligand-bound RTK such as EGFR and FLT3, RAS triggers diverse signaling cascades, including PI3K and MAPK/ERK signaling, to induce cell growth, differentiation, and survival. Oncogenic mutations of RAS in several upstream or downstream pathways occur in most tumors, indicating that inhibition of RAS-dependent signaling is the essential requirement for tumorigenesis. While RAS proteins have been seen as an attractive target for cancer therapy, continuous efforts to inhibit RAS either directly or indirectly through inhibiting post-translational modification or RAS-dependent signaling to date have failed to develop approved therapies for RAS mutant cancer. Previous failures of RAS-targeting treatments, including farnesyltransferase inhibitors, have resulted in RAS being generally considered undruggable. Additionally, because of the high picomolar affinity between RAS and GTP, a competitive inhibitor is not expressly possible. However, the discovery of the structure of RAS has given an understanding of the key residues with new promising compounds. Despite evidence supporting the cancer stem cell theory, resistance to inhibition of RAS appears to be diverse. Durable response to these treatments is yet to be achieved due to complex and diverse mechanisms of adaptive resistance. However, it is expected that employing this rationale will lead to better treatments and a better outcome for patients.

## Figures and Tables

**Figure 1 cancers-13-05599-f001:**
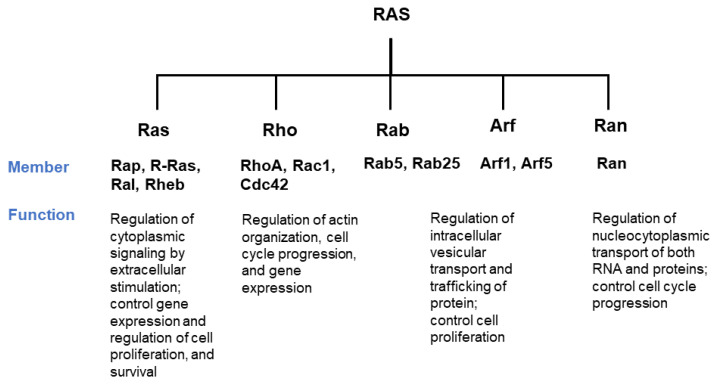
The branched tree of RAS superfamily. RAS superfamily comprises 150 human members and is divided into five different major branches. The Ras sarcoma (Ras) family comprises 36 members and has been the subject of intense research. The Ras homologous (Rho) family comprises 20 members, RhoA, Rac1, and Cdc42 being the best studied. The Ras-like proteins in brain (Rab), the largest branches of the superfamily, comprise 61 members. The ADP-ribosylation factor (Arf) family comprises 27 members; Rab and Arf families have similar functions. The Ras-like nuclear (Ran) family presents only one member [1,8].

**Figure 2 cancers-13-05599-f002:**
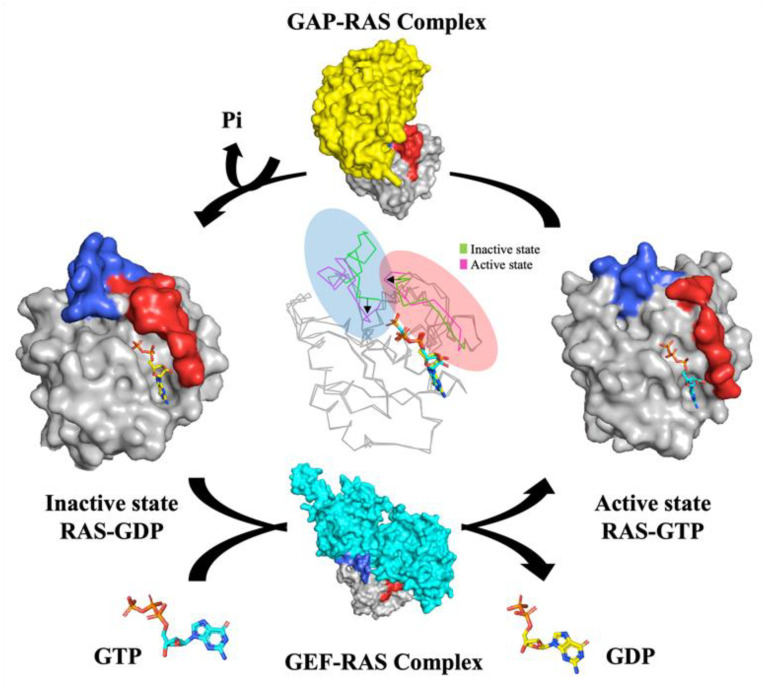
RAS cycles between ‘inactive’ state bound to GDP and ‘active’ state bound to GTP. RAS acts as a guanosine diphosphate (GDP)-inactive molecular switch in resting cells and becomes activated in response to extracellular receptors by binding guanosine triphosphate (GTP), as catalyzed by the guanine nucleotide exchange factor (GEF). The GTPase-activating protein (GAP) stops the Ras signaling by switching Ras into an inactive GDP-bound signaling state. PDB IDs: 7C40 (RAS-GDP), 6Q21 (RAS-GTP), 1WQ1 (GAP-RAS), and 4G0N (GEF-RAS).

**Figure 3 cancers-13-05599-f003:**
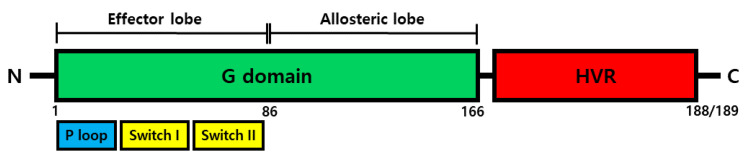
Structure analysis of KRAS. Schematic representation of the full-length KRAS isoform domain. KRAS consists of G domain (residue 1–166) and hypervariable region (HVR, residue 167–188/189). G domain is divided into effector lobe (residue 1–86) and allosteric lobe (residue 87–166). In addition, effector lobe consists of P-loop-phosphate binding loops (residue 10–16), Switch I (residue 30–38), and Switch II (residue 59–76).

**Figure 4 cancers-13-05599-f004:**
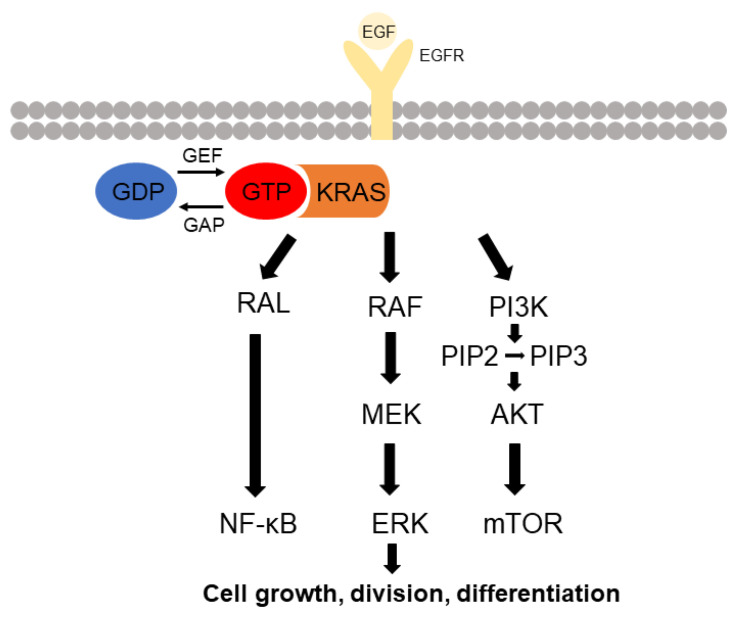
Signaling pathway of KRAS protein. Activation of RAS occurs when epidermal growth factor (EGF) binds to tyrosine kinases receptor such as epidermal growth factor receptor (EGFR). KRAS protein is activated by binding to GTP, transducing the cytoplasmic signal, which activates the RAL, RAF-MEK-ERK, and PI3K-AKT cascade. These cascades regulate cell growth, division, and differentiation.

**Table 1 cancers-13-05599-t001:** Summary of therapeutic strategies towards RAS-driven cancer.

Category	Name	Target	Mode of Inhibition	Ref.
Inhibition of post-translational processing	Cysmethynil	ICMT inhibitor	mislocalization of RAS	[51]
	TLN-4601	MEK/EKR pathway	disrupts RAS membrane anchorage	[52]
	deltarasin	PDEδ	suppresses KRAS localization to endomembrane	[54]
K-RAS targeted drug	ARS-583	KRAS G12C	inhibits downstream MAPK/PI3K signaling	[55]
	ARS-1620	KRAS G12C	anti-tumor activity in subcutaneous xenograft models	[56]
	AMG 510	KRAS G12C	induces pro-inflammatory tumor microenvironment	[57]
	MRTX849	KRAS G12C	tumor regression in KRAS G12C positive cell lines	[58]
	BI-3406	KRAS-mutant of G12 and G13	interacts with the catalytic domain of SOS, resulting in interference with the interaction with KRAS	[59]
	ponatinib and AMG-47a	KRAS G12V	reduces the levels of KRAS G12V proteins	[60]
Inhibition of RAS signaling networks	WZ4002 and MEK inhibitor	EGFR and MEK	effective strategy of drug-resistant cancers and prevents the emergence of drug-resistant clones	[61]
	gefitinib and Src inhibitor	EGFR and Src	potent strategy to overcome Src-mediated ERK reactivation	[62]
	gilteritinib, crenolanib, and midostaurin	FLT3 inhibitor	designed to have a high affinity for the ATP-binding region of the active conformation of FLT3	[63,64,65]
Discovered by virtual screening	BAY-293	KRAS-SOS1 interaction	suppresses the RAS-RAF-MEK-ERK pathway and inhibits the activation of RAS in tumor cells	[66]
	tetrahydro pyridopyrimidines	KRAS G12C	locks KRAS in its inactive GDP-bound form	[67]
	C19	KRAS4B-PDEδ	suppresses the viability and proliferation of colorectal cancer cells	[68]
	D14 and C22	KRAS4B-PDEδ	decreases RAS-GTP activity and ERK and AKT pathways in pancreatic cancer cells	[69]
Other inhibitors	H-REV 107 peptide	KRAS G12V	blocks the activation function of KRAS	[70]
	cisplatin, carboplatin, and oxaliplatin		widely used to treat several cancers	[71]

ICMT: isoprenylcysteine carboxyl methyltransferase, MEK: mitogen-activated protein kinase, ERK: extracellular signal-regulated kinase, PDEδ: phosphodiesterase δ, EGFR: epidermal growth factor receptor, FLT3: FMS-like receptor tyrosine kinase-3.

**Table 2 cancers-13-05599-t002:** Frequency of RAS isoform mutations in human cancer. Table shows the distribution of mutations in the various primary tissue types. Data are from the COSMIC database. Numbers in parentheses indicate sequence of total tested sample.

Primary Tissue	NRAS (%)	KRAS (%)	HRAS (%)
pancreas	0.77 (4016)	52.65 (13,817)	0.11 (3542)
colon	3.87 (16,337)	32.37 (80,177)	0.94 (6095)
small intestine	0.45 (443)	21.34 (1256)	0.3 (336)
lung	0.91 (17,505)	14.19 (45,776)	0.6 (7890)
ovary	1.71 (1932)	13.03 (6932)	0.24 (1693)
liver	1.07 (3174)	3.15 (3614)	0.24 (2912)
kidney	0.34 (3498)	0.93 (4423)	0.25 (3265)
skin	15.36 (15,960)	2.8 (6833)	9.42 (7067)
endometrium	3.22 (1648)	15.45 (4674)	0.58 (2395)
biliary tract	2.7 (2256)	18.05 (5518)	0.56 (1794)

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
