# Peer review of "Oncogenic KRAS: Signaling and Drug Resistance"

_cancers, 2021, doi:10.3390/cancers13225599_

Round 1
Reviewer 1 Report
Authors summarize KRAS protein, KRAS signaling, and RAS inhibitors and resistance in the context of oncogenic KRAS signaling pathway in this review.
Unfortunately, this review is very poorly written and a repetition of recent reviews on the same subject. Authors should consider maybe adding tables of current inhibitors and resistance for different types of cancers instead of already known and published figures of active and inactive state of KRAS cycling.
Please see some of the recently published reviews below:
Tang D, Kroemer G, Kang R. Oncogenic KRAS blockade therapy: renewed enthusiasm and persistent challenges. Mol Cancer. 2021 Oct 4;20(1):128. doi: 10.1186/s12943-021-01422-7. PMID: 34607583.
Fan G, Lou L, Song Z, Zhang X, Xiong XF. Targeting mutated GTPase KRAS in tumor therapies. Eur J Med Chem. 2021 Sep 4;226:113816. doi: 10.1016/j.ejmech.2021.113816. Epub ahead of print. PMID: 34520956.
Author Response
Reviewer1
Comments and Suggestions for Authors
Authors summarize KRAS protein, KRAS signaling, and RAS inhibitors and resistance in the context of oncogenic KRAS signaling pathway in this review.
Unfortunately, this review is very poorly written and a repetition of recent reviews on the same subject. Authors should consider maybe adding tables of current inhibitors and resistance for different types of cancers instead of already known and published figures of active and inactive state of KRAS cycling.
Please see some of the recently published reviews below:
Tang D, Kroemer G, Kang R. Oncogenic KRAS blockade therapy: renewed enthusiasm and persistent challenges. Mol Cancer. 2021 Oct 4;20(1):128. doi: 10.1186/s12943-021-01422-7. PMID: 34607583.
Fan G, Lou L, Song Z, Zhang X, Xiong XF. Targeting mutated GTPase KRAS in tumor therapies. Eur J Med Chem. 2021 Sep 4;226:113816. doi: 10.1016/j.ejmech.2021.113816. Epub ahead of print. PMID: 34520956.
Table 2 has been inserted

Reviewer 2 Report
The author demonstrates the significance of K-RAS mutation in oncogenesis and drug resistance in this manuscript. the Manuscript should be suitable for publication once the following points have been addressed.
- The author should provide detailed information regarding the KRAS superfamily division in lines 38 and 39.
- Lines 41 and 42; The sentence is unclear; it should be rewritten.
- The molecular weight of KRAS protein is 23.2. (Line 45)
- The author should provide the original articles for reference 7
- Line 73, here the author should also provide the information regarding post-translational modification.
- Line 107, Need to improve the sentence.
- For the section 2.1,2.2, and 2.3 (line 123 to 173) author should provide the graphical image to represent K-RAS mediated pathway.
- Reference number 43 (Line 185) is not right (This reference does not show any relation with BMP)
- Reference Number 46, Author should provide original articles do not use review articles.
- Line 219, here it will be great if you start with the heading which represents the K-RAS targeted Drug.
Author Response
Reviewer 2
Comments and Suggestions for Authors
The author demonstrates the significance of K-RAS mutation in oncogenesis and drug resistance in this manuscript. the Manuscript should be suitable for publication once the following points have been addressed.
- The author should provide detailed information regarding the KRAS superfamily division in lines 38 and 39.
[Page 1, Line 34-36] The text of the thesis has been corrected.
- Lines 41 and 42; The sentence is unclear; it should be rewritten.
[Page 1, Line 37-39] The text of the thesis has been corrected.
- The molecular weight of KRAS protein is 23.2. (Line 45)
[Page 1, Line 41-42] The text of the thesis has been corrected.
- The author should provide the original articles for reference 7
[Page 1, Line 41] The text of the thesis has been corrected.
- Line 73, here the author should also provide the information regarding post-translational modification.
[Page 2, Line 70-75] The text of the thesis has been corrected.
- Line 107, Need to improve the sentence
[Page 4, Line 118-119]. The text of the thesis has been corrected.
- For the section 2.1,2.2, and 2.3 (line 123 to 173) author should provide the graphical image to represent K-RAS mediated pathway.
[Page 5, Line 163-168], The figure has been added to the body of the article.
- Reference number 43 (Line 185) is not right (This reference does not show any relation with BMP)
Reference 43. The text of the thesis has been corrected.
- Reference Number 46, Author should provide original articles do not use review articles.
Reference 68. The text of the thesis has been corrected.
- Line 219, here it will be great if you start with the heading which represents the K-RAS targeted Drug.
The title of line 252 has been corrected.

Reviewer 3 Report
The review is well written. Few changes are recommended.
- This section ". The Ras superfamily is divided into sequence and functional that this family is comprised of RAS, RHO, RAB, RAC, RAN,
and ARF. The RAS family is divided into 6 subfamilies (RAS, RAL, RAP, RHEB, RAD, and RIT" may be shown as branched tree with some details of the function of each of the member. - RAS history can be either omitted or elaborated. Readers will not gain any information from the current content.
- Figure 2 can be shown in ribbon form with XYZ co-ordinates to better understand the structural differences between GTP bound and GDP bound forms.
- The Ral-GEF protein is inadequately represented. Please add some details with respect to colorectal and lung cancer.
- Please provide a reference for this statement " Resistance mechanisms happened in a very rare subpopulation of cells, such as cancer stem cells.
- Please add few sentences as to why KRAS has been a non-viable (undruggable) therapeutic target.
Author Response
Reviewer 3
Comments and Suggestions for Authors
The review is well written. Few changes are recommended.
- This section ". The Ras superfamily is divided into sequence and functional that this family is comprised of RAS, RHO, RAB, RAC, RAN,
and ARF. The RAS family is divided into 6 subfamilies (RAS, RAL, RAP, RHEB, RAD, and RIT" may be shown as branched tree with some details of the function of each of the member.
[Page 2, Line 59-65], The figure has been added to the body of the article.
- RAS history can be either omitted or elaborated. Readers will not gain any information from the current content.
Content has been added and modified to help readers understand.
- Figure 2 can be shown in ribbon form with XYZ co-ordinates to better understand the structural differences between GTP bound and GDP bound forms.
[Page 6, Line 211-212] The text of the thesis has been corrected.
- The Ral-GEF protein is inadequately represented. Please add some details with respect to colorectal and lung cancer.
[Page 6, Line 214-215] The text of the thesis has been corrected.
- Please provide a reference for this statement " Resistance mechanisms happened in a very rare subpopulation of cells, such as cancer stem cells.
References 40 and 41 were added to the body of the paper.
- Please add few sentences as to why KRAS has been a non-viable (undruggable) therapeutic target.
Added content to the thesis Conclusion.

Round 2
Reviewer 1 Report
Authors added tables of current inhibitors and resistance for different types of cancers and improved the manuscript.
Author Response
Reviewer1
Comments and Suggestions for Authors
Authors added tables of current inhibitors and resistance for different types of cancers and improved the manuscript.
Thanks for your comment.

Reviewer 2 Report
The manuscript is well written and well represented. I will be happy if the following suggestion should be included.
Cancer stem cells play a significant role in cancer recurrence following surgery, and the K-RAS mutation is critical in the generation of cancer stem cells. I feel the author should discuss the role of K-RAS in CSC. The majority of K-ras mutant cells exhibit enhanced autophagy which induces the cancer cell stemness under stress conditions which potentiates drug resistance (PMID 33946505).
Author Response
Reviewer 2
Comments and Suggestions for Authors
The manuscript is well written and well represented. I will be happy if the following suggestion should be included.
Cancer stem cells play a significant role in cancer recurrence following surgery, and the K-RAS mutation is critical in the generation of cancer stem cells. I feel the author should discuss the role of K-RAS in CSC. The majority of K-ras mutant cells exhibit enhanced autophagy which induces the cancer cell stemness under stress conditions which potentiates drug resistance (PMID 33946505).
Reference 43 has been added.
